# Molecular Challenges and Opportunities in Climate Change-Induced Kidney Diseases

**DOI:** 10.3390/biom14030251

**Published:** 2024-02-21

**Authors:** Eder Luna-Cerón, Alfredo Pherez-Farah, Indumathi Krishnan-Sivadoss, Carlos Enrique Guerrero-Beltrán

**Affiliations:** Tecnologico de Monterrey, Escuela de Medicina y Ciencias de la Salud, Monterrey 64710, Nuevo León, Mexico; elunacer@ttuhsc.edu (E.L.-C.); alfredo.pherezfarah@studenti.unipd.it (A.P.-F.); indumathikrishnan1@gmail.com (I.K.-S.)

**Keywords:** heat stress, heat shock proteins, sirtuins, chronic kidney disease, RAAS, fructose uptake, therapeutic, molecular mechanisms, public health

## Abstract

As temperatures continue to modify due to weather changes, more regions are being exposed to extreme heat and cold. Physiological distress due to low and high temperatures can affect the heart, blood vessels, liver, and especially, the kidneys. Dehydration causes impaired cell function and heat itself triggers cellular stress. The decline in circulating plasma volume by sweat, which stresses the renal and cardiovascular systems, has been related to some molecules that are crucial players in preventing or provoking cellular damage. Hypovolemia and blood redistribution to cutaneous blood vessels reduce perfusion to the kidney triggering the activation of the renin–angiotensin–aldosterone system. In this review, we expose a deeper understanding of the modulation of molecules that interact with other proteins in humans to provide significant findings in the context of extreme heat and cold environments and renal damage reversal. We focus on the molecular changes exerted by temperature and dehydration in the renal system as both parameters are heavily implicated by weather change (e.g., vasopressin-induced fructose uptake, fructogenesis, and hypertension). We also discuss the compensatory mechanisms activated under extreme temperatures that can exert further kidney injury. To finalize, we place special emphasis on the renal mechanisms of protection against temperature extremes, focusing on two important protein groups: heat shock proteins and sirtuins.

## 1. Introduction

Climate change is causing a series of changes in meteorological parameters that in the last century have manifested as extreme weather conditions and temperature variability. A global rise of 1 °C in mean temperature has brought serious health hazards and no region seems to be spared [1]. Although a clear international consensus linking weather change and human diseases has not been clearly established to implement policies and preventive measures against this global phenomenon, a surplus amount of quality evidence has been published in the last decade showcasing cardiovascular and renal health complications. Extreme temperatures can induce organ damage and increase the susceptibility for the development of cardiovascular and renal diseases as both participate in thermoregulatory mechanisms. In cardiovascular diseases, several studies have indicated links with extreme temperatures on both ends of the spectrum: high temperatures and cold spells [2,3,4]. Research indicates that cold spells increase cardiovascular emergency hospitalization rates [5,6]. Other studies show that high temperatures are also associated with a variety of cardiovascular diseases such as acute myocardial infarction, arrhythmias, and heart failure [7]. Although both hot and cold spells contribute as dangerous risk factors for cardiovascular diseases, some experts highlight that cold spells have a prolonged effect and a higher mortality impact versus hot spells [2,4]. These studies bring insight to a rising public health concern as many geographical regions and countries of different socio-economic backgrounds are increasingly exposed to temperature variations due to the climate crisis. Given the current epidemiological importance of kidney disease, the following discussion focuses on the chronic impact on kidneys due to prolonged exposure to high temperatures, shedding light on the intricate relationship between temperature variations and kidney diseases.

## 2. Kidney Health in Extreme Temperature Conditions

Focusing now on the kidneys, there is evidence that chronic activation of counterregulatory mechanisms against hyperthermia can lead to processes that exert further kidney injury [8]. Clinical studies have reported a higher incidence of urinary and kidney diseases such as urinary tract infections, chronic kidney disease (CKD), and acute kidney injury (AKI), when people are exposed to high temperatures, where heat waves seem to be important triggering factors in recent years [9,10,11]. In the past two decades, numerous studies conducted among young agricultural workers in warm regions of Central America have consistently documented increasing prevalence rates of CKD compared to other regions of the world [12,13,14,15]. As a result, the term Mesoamerican nephropathy was coined, for which recurrent dehydration and volume depletion were identified as possible mechanisms of this type of renal lesion [16].

In other tropical and subtropical regions, such as in India and Southeast Asia, higher rates of CKD have been reported without the identification of clear risk factors such as hypertension and diabetes [17]. This has been denominated CKD of uncertain etiology (CKDu) and has been associated with Mesoamerican nephropathy as both pathologies seem to share similarities, mainly continuous renal damage due to subclinical increases in temperature [18]. Indeed, recent clinical data from both regions suggests these two terms might refer to the same clinical entity [19,20,21,22]. This pathology is now recognized as a well-established occupational disease and a growing health concern in regions such as Asia and Latin America. As its name implies, its pathophysiology is not well known and its exact source is uncertain; however, it is widely accepted that heat stress, dehydration, strenuous labor, use of agrochemicals, exposure to heavy metals, and water pollution are contributing factors or, as Priyadarshani et al. (2022) have mentioned, as “suspected causative factors” for this pathology [23]. It is not surprising that mine and farm workers present a greater decrease in their kidney function due to their labor conditions [24,25,26]; nevertheless, causality cannot be assigned to only one or two factors, as was mentioned before, without taking into consideration the stress conditions given to the individual at a certain moment in their occupational labor.

Exposure to high ambient temperature not only signifies a risk of developing CKDu, but also increases kidney-related hospitalizations, which turns it into a public health concern as well [27,28]. In fact, a recent meta-analysis of 82 studies found that with a 1 °C rise in temperature, kidney-related morbidity increased by 1% [29]. This finding is in line with a recent global analysis in which it was concluded that age-standardized mortality rates and age-standardized rates of disability-adjusted life years due to high temperatures have been consistently increasing since 1990, especially in tropical regions [30].

Many studies stress the notion that implications of hot and cold spells preferentially affect low-income nations located in tropical regions. However, high-income countries such as the United States, parts of Europe, and Australia also demonstrate reduction in work capacity and performance due to heat, as well as increased hospitalization rates and death tolls [31,32,33,34]. For example, data gathered in a 12-year period in South Australia demonstrated a 10% increase in hospital admissions due to renal disease during heat waves associated with higher AKI rates [35]. In Adelaide, Australia, other studies have reported higher hospitalization rates for renal disease as well [36,37]. In New York and Atlanta, similar reports were found [38,39]. Rising temperatures seem to predispose residents to urolithiasis as well, with warmer states in the United States posing a greater risk for stone formation [40]. As a result of climate change, the already established U.S. kidney stone “belt” in southeast regions of the country seems to be expanding northward [40]. Increases in mean daily temperatures were also associated with higher outpatient urinary tract infection cases among women in a case-crossover report in California [41]. These studies demonstrate seasonality in kidney diseases, putting forth warmer climates as a potential risk factor.

Although heavily studied risk groups include mine, agriculture, and construction workers due to occupational heat exposure, our aging population has greater risk factors for renal failure as changes in body composition, reduced thirst mechanisms, and greater fluid losses add propensity to dehydration [42]. Studies indicate that pediatric patients are at greater risk as well for renal disease during heat waves [36,43]. Air-conditioning might not be an effective solution as it makes humans more sensitive to heat exposure due to lack of acclimatization [31]. Although there is a wealth of evidence demonstrating the harmful effects of heat exposure on kidneys, evidence is lacking at the other end of the spectrum in colder climates. Whereas in the cardiovascular system, it is established that both extremes of the scale prove to be harmful for humans.

As climate models predict frequent and more severe heat waves and cold spells in coming years, it is important to start understanding and integrating possible molecular mechanisms that cause kidney injury in high and low temperatures, as well as key proteins that aid in cellular homeostasis under extreme temperature conditions. In this review, we will focus on the molecular changes exerted by temperature in the renal system as some parameters are heavily implicated in kidney diseases. A deeper understanding of the modulation of molecules that interact with other proteins in humans could provide significant findings in the context of climate change and renal damage reversal.

## 3. Compensatory Mechanisms Activated under Extreme Temperatures Can Exert Further Kidney Injury

### 3.1. Mechanisms and Biomarkers Associated with Kidney Damage Due to High and Low Temperatures

#### 3.1.1. Vasopressin, Fructose Uptake, Osmotic Damage, and Inflammation

Acute or chronic exposure to high temperatures leads to insensible volume losses, which in turn stimulate different compensatory cardiovascular and endocrine responses as well as kidney injury produced by different mechanisms. For example, the release of vasopressin from the neurohypophysis, which increases water and urea reabsorption in the collector tubes of nephrons, promotes reduction in urinary volume and urine concentration [44]. Although these mechanisms are intended to maintain adequate arterial pressure, low urine outputs and higher urine concentration can lead to the accumulation of crystals and urinary lithiasis development [45]. Consistently, epidemiological data suggest a strong correlation between temperature and kidney stone formation, and this relationship is more significant in higher temperatures [46]. Shifting the focus on beneficial mechanisms that become deleterious in chronic settings, there are two pathways proposed to be related to renal injury that activate during high temperatures and dehydration: vasopressin release and fructose metabolism. Both of these mechanisms may seem useful in protecting against volume loss; however, studies have demonstrated that their chronic activation is linked to CKD [47].

Animal models exposed continuously to high temperatures have shown that an increase in serum osmolarity and the consequent response with higher vasopressin levels can induce fructose uptake from cells of the proximal convoluted tubule and induce fructogenesis [48]. This is achieved by enhancing the activity of the rate-limiting enzyme aldose reductase, which favors the conversion of glucose to sorbitol, which is then transformed to fructose by sorbitol dehydrogenase [49,50,51]. In this sense, sorbitol can induce cell dysfunction by mediating proximal convoluted tubule cell osmotic damage. Consistently, there is clear evidence that fructose can induce AKI development and that repeated AKI events can eventually lead to the development of CKD [52].

Clinical studies have reported that pediatric patients undergoing surgery diagnosed with AKI had higher fructose concentrations in the urine [53]. This data was confirmed in animal models of kidney ischemia–reperfusion that showed severe tubular damage compared to knockout animals for key enzymes of the fructogenesis pathway [54]. These important pathophysiological hallmarks were also found in animal models of chronic and recurrent dehydration [55]. Several explanations for the deleterious effects of fructose are centered in the fact that fructokinase, a key enzyme in the catabolism of fructose, requires the consumption of inorganic phosphate, favoring the ATP depletion and the catabolism of AMP, thus producing an increase in urate as a product of this pathway, especially in situations with high fructose uptake as seen in patients with chronic dehydration [55,56].

Elevated urate and uric acid levels can then induce further glomerular and tubular damage by compromising the production of nitric oxide, favoring the synthesis of reactive oxygen species (ROS) and promoting endothelial damage [57]. Moreover, the higher release of uric acid, which occurs in the context of heat induced by exercise, tends to concentrate and crystallize, increasing the urolithiasis risk [58]. It is important to know that in the context of heat-induced dehydration in agricultural workers, it is more likely that toxic agents such as heavy metals, pesticides, and tobacco-related chemicals can concentrate in the tubular lumen of nephrons, making them more susceptible to direct damage [59]. As a consequence of the mechanisms explained, it is clear that fructose metabolism can induce the production of detrimental metabolites that exert oxidative stress, inflammation, and increase the release of uric acid, which in turn generate more local damage [49,60].

In addition to the effects of fructose, AKI is a state characterized by the presence of hypoperfusion and hypoxia [61]. In dehydration, reductions in the cardiac output impact the renal blood flow and thus the glomerular filtration, leading to reduced delivery of nutrients to renal tubular cells. This state generates tubular cell inflammation and necrosis [61,62]. Damaged tubular cells produce damage-associated molecular patterns (DAMPs), which are then recognized by immune and non-immune cells through several pattern recognition receptors, such as the Toll-like receptors, thus producing several proinflammatory mediators such as interferon-γ and interleukin-1β [63,64]. In this sense, nuclear factor kappa-B (NF-*k*B) plays an important role in regulating inflammatory-mediated damage, as seen in preclinical models where inhibition of NF-*k*B reduces tubulointerstitial damage. In this regard, fructose, fructokinase activation, and some nephrotoxins have been associated with the upregulation of NF-*k*B, which is a potent transcriptional inductor of genetic sequences associated with inflammatory mediators such as interleukin-1β, interleukin-6, and macrophage chemoattractant molecule-1, which are well known to enhance the production of ROS directly by leukocytes such as macrophages or neutrophils but also by the intrinsic production of these molecules inside tubular cells [65,66]. These mechanisms are summarized in Figure 1.

Furthermore, in addition to the compensatory mechanisms linked to volume deprivation during high-temperature exposure, the proper functioning of both the renin–angiotensin–aldosterone system (RAAS) and the sympathetic nervous system have been compromised under adverse temperature conditions, as elucidated in the subsequent section.

#### 3.1.2. Renin–Angiotensin–Aldosterone System

In addition to the well-known etiology of hypertension, it has been established that a seasonal variation also exists for this condition. Various studies have reported higher recorded blood pressures in colder temperatures versus warmer climates [67,68,69,70]. The possible mechanisms behind this phenomenon have been associated with the renin–angiotensin–aldosterone system (RAAS) and the sympathetic nervous system. Studies have demonstrated that an acute exposure to cold temperatures causes a rise in plasma norepinephrine, elevating the blood pressure [71,72]. Furthermore, studies indicate that this stimulation comes from the central nervous system and not from the adrenal glands, as demonstrated through the observation of a rise in blood pressure in response to cold in adrenalectomized patients [73,74]. While the role of norepinephrine in elevating blood pressure is well established in cold temperatures, the involvement of the RAAS system is controversial in the same setting.

Some reports have demonstrated the elevation of renin, angiotensin II, and/or aldosterone after cold exposure [75,76], whereas others have reported a null response [71,77]. There are, however, studies that indicate a more prominent role of RAAS under these circumstances. For example, in patients after cold immersion, the administration of the angiotensin-converting enzyme inhibitor, captopril, lowered plasma norepinephrine levels and increased plasma renin levels indicating an effective blockade of the angiotensin-converting enzyme and a negative effect in sympathetic activity [78]. The results demonstrate that both systems interact with one another under cold temperatures, with an increase in blood pressure as a result. While these reports have primarily concentrated on cardiovascular diseases to establish a correlation with cardiovascular mortality and RAAS system alterations, there is a notable absence of studies investigating the long-term impact of winter-associated blood pressure elevation on CKD, limiting the research of the effects of low temperatures on the development of CKD. Exploring the chronic effects of these mechanisms on the kidneys could be intriguing, particularly considering that hypertension stands as a predominant cause of CKD globally.

Regarding the role of the RAAS system in high temperatures, the evidence is also scarce and controversial. It has been long observed that plasma renin, angiotensin II, and aldosterone activity rise upon environmental heat stress, which can be explained by increased renin production possibly due to decreased renal blood flow, reduced catabolism due to decreased hepatic blood flow, and RAAS stimulation due to catecholamine release [79,80]. Indeed, a study conducted by Eisman and colleagues in environmentally heat-stressed baboons showed that as the temperature rose, renal blood flow decreased, whereas renal vascular resistance and plasma renin activity (PRA) increased in a linear fashion. These effects were partially abrogated by propranolol, α β-blocking agent, and saralasin, an angiotensin-II antagonist; suggesting that hyperthermia-induced renal vasoconstriction is partly explained by a β-adrenergic-mediated release of renin [81]. This last statement is supported by the fact that, in humans, a pharmacologic β-blockade significantly reduces heat-induced PRA [82].

It is worth mentioning that an increase in baseline levels of different components of the RAAS system has been observed by multiple groups after heat acclimatization, suggesting the possibility of a chronically activated cascade [83,84,85]. Nonetheless, it is worth mentioning that other groups did not observe such a relationship, highlighting the interplay of many different variables including environmental factors, hydroelectrolytic status, and the state of other cardiovascular parameters in this context [86,87].

To understand the fundamental mechanisms related to RAAS modulation and its effects on the cardiovascular system, a detailed introduction can be found in the following reviews [88,89]. The fact that the interdependency of sympathetic activation and the RAAS system is found on both sides of extreme temperature settings poses the idea that it is extreme changes in climate, rather than a specific temperature range, that drives these detrimental effects on kidney physiology.

The constant activation of the pathways described brings disarray and causes kidney injury. When analyzing the effects of changing temperatures, the chain of reactions elucidated in the sections above shows to be a potential link between the neurohumoral response exerted primarily by vasopressin and metabolic dysregulation in fructose metabolism and hot/cold spells. On the other hand, when focusing on the effects of low temperatures, in certain conditions RAAS and the sympathetic system are stimulated resulting in a raise in blood pressure. Nevertheless, there are beneficial nephroprotective biomarkers, notably sirtuins (SIRTs) and heat shock proteins (HSPs), which possess the potential of alleviating temperature-induced kidney damage.

## 4. Heat Shock Proteins and Sirtuins as Renal Biomarkers of Protection against Extreme Temperatures

High temperatures and dehydration might increase cellular resistance to various stress forms through mechanisms that are well-known, e.g., HSPs and SIRTs, which are abundant cellular proteins involved with protein homeostasis; they are involved in the relationship between energy balance and gene transcription, allowing cells to respond to caloric restriction and to survive conditions with oxidative stress such as temperature elevation, reduced oxygen levels, infection, inflammation, and exposure to toxic substances. In this section, we will elucidate and summarize various beneficial roles undertaken by these proteins under stress conditions, specifically, under heat stress.

### 4.1. Heat Shock Proteins

As priorly discussed, there is evidence from both animal and human studies that shows that exposure to high temperatures can induce cellular stress in several organs including the kidneys [20,90,91]. In this regard, stress is one of the principal factors that trigger protein unfolding, misfolding and/or aggregation, which causes a response that mainly leads to the induction of certain gene transcriptions of proteins [92]. Interestingly, one of the compensatory mechanisms of the kidney to provide protection against cellular stress induced by high temperature exposure is elicited by heat shock proteins (HSPs). HSPs are a small family of chaperones found in different types of organisms ranging between bacteria, yeast, and humans [93]. The better characterized HSPs are named according to their mass in kilodaltons, such as: HSP90, HSP70, HSP60/65, HSP32, and HSP27. Several HSPs contribute to reducing the aggregation of proteins damaged by diverse causes, including heat. For this reason, cellular levels of HSPs may increase in heatstroke conditions (as previously discussed) while remaining low in the absence of stressors. Therefore, HSPs play a major role in the biological response to stress, and its modulation and understanding is of interest in diverse pathologies [93,94].

This intrinsic involvement in stress response becomes particularly relevant when considering the broader environmental changes associated with global warming. The escalating environmental temperatures, explained in the context of global warming, have the potential to induce chronic overexpression of HSP genes, triggered by both heat and cold stress [95]. Therefore, the initial acknowledgment of HSPs’ importance in stress response provides a foundation for comprehending their adaptive functions, which, when extended to the implications of global warming, demonstrates their intricate role in mitigating the impact of extreme environmental conditions on cellular health. In this regard, it has been suggested that this variation in HSP expression might be a cellular adaptation response aimed to cope with diverse adverse temperature conditions and environment change [95,96].

HSP expression has been associated with both beneficial and deleterious effects, depending on the specific HSP phenotype [93]. As previous authors have reported, HSPs can enhance thermal stress tolerance and thermal plasticity but can also result in cell cycle arrest and cell death [97,98,99]. In the renal system, these proteins have been widely characterized in different pathological conditions within specific sites of the nephron [100]. In consideration of the diverse range of protective effects against cellular stress and the corresponding adverse impacts on cell survival, we emphasize the necessity for further research to comprehensively elucidate the exact role of HSPs in the specific context of exposure to elevated temperatures. Having better evidence can foster a clearer understanding of the implications of HSP expression within the broader framework of global warming [95].

The subsequent sections provide a more comprehensive elucidation of the specific roles played by distinct Heat Shock Proteins (HSPs) in safeguarding against kidney injuries induced by exposure to adverse temperatures.

#### 4.1.1. Heat Shock Protein 27

From this complex family, HSP27 plays a critical role in the protection of the kidney against adverse environments. This protein family is highly expressed in the proximal convoluted tubule and collector tubular cells in the kidney medulla [101,102]. However, this family of HSP has been reported to also be expressed within glomerular capillaries [103]. Within its wide set of actions in the context of AKI, HSP27 is capable of stabilizing actin polymers in circumstances of stress (such as the presence of ROS or tumor necrosis factor alpha). Furthermore, it is also known to protect against osmotic damage and to inhibit the activation of the intrinsic apoptotic pathway [104]. The expression of these proteins has not only been present in cases of high temperature injury, but is also associated with the effects of nephrotoxic agents such as cyclophilin [105]. At the glomerulus, HSP27 has also been involved in the maintenance of podocyte permeability [102,106]. Accordingly, preclinical models of nephrotic syndrome have shown an increase in the aggregation and phosphorylation of HSP27, possibly as a compensatory mechanism to maintain the filtration barrier of podocyte cytoplasmic extensions [102].

In a similar fashion, HSP27 has been reported to restore partially damaged proteins and to maintain the integrity of the cytoskeleton [107]. Certainly, phosphorylation of this HSP family regulates the interaction with target proteins to facilitate their protective effects. For example, in the phosphorylated state of HSP27, its structure and intracellular distribution are altered, leading to cellular growth and survival. When HSP27 is dephosphorylated, it exists in small structures on the cytoplasm leading to less interaction with target proteins. In states of overactivation of oxidative stress mediators or inflammation, which are known to be associated with heatstroke, HSP27 is able to protect against ROS, and it also modulates cellular glutathione and protects the cells from tumor necrosis factor alpha injury [108]. Given the importance of the effects of HSP27 and its activation through phosphorylation switches, it can be a possible therapeutic target and biomarker for both acute and chronic renal injury in the context of heatstroke.

#### 4.1.2. Heat Shock Protein 70

Another chaperone member, the HSP70 proteins, have also seen an increase in preclinical models of toxic and ischemic AKI [109]. These chaperones play a crucial role in cellular thermotolerance, given that they prevent the aggregation of proteins that have been denatured by heat stress. Due to their cytoprotective function and role in thermal adaptation, higher levels of HSP70 have been found during conditions of thermal stress [110,111]. However, extreme ambient temperature can cause post-translational regulation through CDK-dependent HSP70 phosphorylation, which may in turn result in changes in chaperone interactions with cyclins and alter cell cycle progression [112]. Moreover, in conditions of increased environmental stress, such as that caused by limited water supply and dehydration states, HSP70 mRNA post-transcriptional methylation modifications can occur and lead to changes in HSP70 translation. Specifically, selective translation of HSP70 may take place, resulting in a suppression of global translation and a reduced effectiveness of HSP70 in the heat shock response [113]. In this line, in another study it was suggested that HSP72 (a member of this family) has been shown to help in the adequate folding and function preservation of Na^+^-K^+^-ATPase and, thus, the osmotic balance of damaged renal cells [114]. Possibly, in chronic dehydration conditions, the overall expression of HSP70 is reduced and, thus, their effects on the osmotic balance are lost, leading to further renal injury.

Consistent with these results, animal models of ischemic-induced AKI have shown higher levels of HSP70 protein expression in comparison with healthy controls. In this sense, this family of proteins has been described to have adequate sensitivity as biomarkers of AKI instauration and recovery [115,116]. In addition to their role in protein integrity, these molecules have been shown to downregulate the activity of pro-apoptotic proteins such as bcl-2-like protein 4 and Caspase 3 [117]. In this same study, HSP70 proteins have been involved in the inhibition of the nuclear translocation of NF-*k*B p65 subunit, therefore reducing the pro-inflammatory signals produced by damaged tubular cells in AKI [114]. Additionally, these proteins seem to promote benefits in chronic consequences of renal damage (such as fibrotic scar formation) by downregulating the activity of transforming growth factor beta and type 1 collagen, which are key mediators of fibrogenesis [118]. Furthermore, HSP70 proteins have been reported to increase the activity of key kinase proteins such as the mitogen-activated protein kinases, thus promoting the expression of genes involved in cell proliferation and differentiation [119].

#### 4.1.3. Heat Shock Protein 90

HSP90 is an interesting chaperone since it collaborates with other members of this family. Some reports have shown the activity of the HSP90 family to promote the ubiquitination and destruction of the receptor for transforming growth factor beta [120,121], a similar effect has previously described by HSP70. HSP90 interacts with steroid and hormone receptors and participates in intracellular signaling. In conditions of high ambient temperatures, the expression of HSP90 in the kidney, heart, and central nervous system increases [122]. In fact, the expression of both HSP70 and HSP90 was causally associated with dehydration tolerance [123]. Nonetheless, the mechanisms by which HSP90 exerts renal protection remain scarcely studied.

In this regard, a recently published study conducted on an in vivo mouse model of heat stress demonstrated that HSP90 limited AKI by upregulation of the PKM2-Akt anti-apoptotic and the HIF-1α-BNIP3/BNIP3L autophagy-related pathways. Interestingly, PKM2-Akt signaling is also known to activate HSP70. This evidence suggests that HSP90 mainly favors tubular cell survival by the inhibition of key apoptotic mediators and by the induction of autophagy and denotes a new insight of cross-talk between different HSPs [124].

Moreover, following the association between renal stone formation and dehydration induced by heatstroke, HSP90 has been shown to upregulate the expression and binding capacity of calcium-oxalate binding proteins such as beta actin, vimentin, and calpain-1, thus inhibiting the intratubular precipitation of crystal and stone formation [125]. In this sense, a possible link between dehydration and HSP90 upregulation can be the RAAS, which is constitutively activated in this clinical setting, as aldosterone has previously been reported to induce the expression of HSP90 in an in vivo model [126]. However, inhibition of HSP90 has also been exhibited to improve overall renal function in other contexts by downregulation of inflammatory responses such as NF-*k*B nuclear translocation or Toll-like receptor 4 activation in mouse animal models of ischemia [127].

In the clinical field, HSP90 has been described to increase in the presence of AKI, especially when associated with drug-induced toxicity [128]. Overall, this information indicates that HSP90 can counteract the deleterious effects of heatstroke, mainly by increasing tubular cell survival and inhibiting the precipitation of crystals or toxic agents, especially in the context of dehydration.

#### 4.1.4. Heat Shock Protein 60

HSP60 has also been associated with exerting protective effects within the nephron that can play a critical role in the defense of heatstroke. This protein family has been reported to be expressed predominantly within the renal cortex and in the external boundaries of renal medulla [129]. Interestingly, the distribution of this chaperone mimics the abundance of mitochondria within the kidney [130]. This anatomical distribution is critical in the context of heat-induced kidney injury since the most affected renal region during periods of hypoperfusion and hypovolemia is the transition at the corticomedullary region [131]. Overall, HSP60 expression has been shown to increase in heat stress conditions, but since it is barely expressed in the renal inner medulla, it has not been associated with hyperosmotic damage [132].

In a study conducted in heat stress-induced renal damage on chicken models by Tang et al. (2018), it was shown that after exposition to high temperatures in broilers, the expression of HSP60 increased significantly after 1 h of damage [133]. Of note, these animals also have compatible features of AKI, such as increased levels of urea and uric acid, while the administration of aspirin reduced the clinical and histopathological hallmarks of damage and maintained a constitutive expression of HSP60 [133]. This evidence suggests that protective pathways against heat stroke-induced AKI involve HSP60-dependent pathways. Moreover, a previous study with a rat model of dehydration showed that, after three days of water restriction, HSP60 expression increased in the mitochondria of papillary cells, mainly by importation from the cytosol [134].

Additionally, with their effects in hypoperfusion, HSP60 proteins have also been recognized to protect against the accumulation of heavy metals. For example, in a rat model of acute tubular necrosis induced by inorganic mercury, the administration of this agent was associated with a higher magnitude of renal injury and expression of HSP60, which coexisted with the histological features of necrosis, especially at the levels of cortical tubules. Importantly, in this same study, the expression of HSP60 was higher at the mitochondria and cytoskeleton of injured tubular cells compared to controls. Similarly (in a rat model), increases in bioavailability of selenium induced a higher expression of HSP60 [135].

The integration of these results suggests that HSP60 may play a critical role in the refolding and repair of mitochondrial proteins in stress conditions (due to heavy metal toxicity or hypoxia), reflecting the importance of this chaperone in sites of high energy consumption such as the corticomedullary tubular cells. Even more, there is clinical evidence that HSP60 can be used as a biomarker in pediatric patients with sepsis and recently developed AKI [136]. Further research is needed to clarify the molecular mechanisms involving HSP60 and its expression in clinical studies of heat stress-induced AKI.

#### 4.1.5. Heat Shock Protein 32

Finally, HSP32 or heme-oxygenase (HO-1), which is an enzyme whose products, CO and biliverdin, are potent ROS scavengers, has also been shown to increase in the context of certain stressors such as increased temperatures, heavy metal intoxication, and ROS exposure, which (as described previously) are key drivers of AKI in the context of temperature stress [137]. In later stages of kidney damage, some leukocytes are polarized into anti-inflammatory lineages (such as the type M2 macrophages), which induce the cessation of the inflammatory milieu by the direct action of anti-inflammatory cytokines such as interleukin-10 or by the expression of antioxidant mediators such as HSP32 [138,139,140]. In this regard, HSP32 has been previously described to reduce the levels of proinflammatory cytokines (such as interleukins-1, -6, and -8) and reduce the effects of ROS as an antioxidant agent [141,142]. In healthy kidneys, their main expression of this protein has been located on the proximal convoluted tubule and Mesangium [143]. The levels of HSP32 increase as a result of several mechanisms of kidney damage such as ischemia, toxic injury, and diabetic nephropathy.

With respect to their immunomodulatory actions in kidney injury, HSP32 has been reported to be expressed mainly by T regulatory lymphocytes [144]. Even clinical studies have demonstrated that low levels of HSP32 expression show a negative correlation with the risk of development of mesangioproliferative glomerulonephritis [145]. Additionally, HSP32 has shown antifibrotic properties by downregulating transforming growth factor beta [146,147]. These results suggest that HSP32 mainly plays a crucial role in the modulation of the inflammatory response induced by AKI and that its effects are synergistic with those of HSP70 and HSP90 families [148,149].

The most important HSPs expressed in AKI, the pathological state associated with heatstroke, are summarized in Table 1.

### 4.2. Sirtuins

Sirtuins (SIRTs) constitute a protein family that plays pivotal roles in maintaining homeostasis under various stressors. This evolutionarily highly conserved protein family exhibits either mono-ADP-ribosyltransferase or NAD^+^-dependent deacetylase activity. Additionally, they are recognized for exerting some of their actions through deacetylase-independent mechanisms [150]. Seven SIRTs (1–7) have been identified in mammals, each of which participates in the regulation of several molecular and biochemical processes, including energy homeostasis, cell proliferation, DNA replication and repair, protein synthesis and modification, inflammation, oxidative metabolism, and cell senescence and death. Although most experimental evidence highlights their overwhelmingly protective role in the kidneys, their great functional diversity makes them a double-edged sword as their cellular pathways closely rely on the underlying physiological or pathophysiological circumstances which condition their final effect.

A link between SIRT metabolism and heat-associated nephropathies might seem unlikely at first glance, but it becomes quite plausible when considering that kidneys are high-energy-demanding organs, essential for thermic adaptation, and SIRTs are very efficient stress sensors, essential for proper kidney function [151]. Most studies regarding their thermodynamic regulation have been conducted in invertebrates and fish, exposing a greatly unexplored area of research in human beings [152].

#### 4.2.1. Sirtuins and RAAS

The most evident connection between heat-stress and kidney disease lies in heat-induced dehydration, which leads to a series of compensatory mechanisms such as osmoreceptor stimulation, adrenergic response, vasopressin release, and RAAS activation (Figure 2).

Even though these mechanisms intend to maintain hydroelectric homeostasis, they can be harmful on the kidneys if left unregulated. For instance, in the collecting duct, SIRT1 possess the ability to induce aquaporin-2 expression via NF-*k*B inhibition and repress the epithelial sodium channel transcription through a deacetylase-independent mechanism, hence regulating sodium reabsorption whilst favoring water transport, potentially mitigating the overall damage of RAAS overactivation by modulating the effect of aldosterone and reducing renin stimuli, respectively [153,154]. Heat can also alter acid-base homeostasis, especially in the presence of predisposing medical conditions or genetic variants prevalent in certain geographical regions, such as in Thailand, where distal renal tubular acidosis is an endemic condition, especially during the summer [155,156]. Recently, a report found that the renal cotransporter potassium-chloride cotransporter 4 (essential for renal tubular acidosis compensation) is stabilized and enhanced by SIRT7 due to a deacetylase-dependent ubiquitination inhibition, protecting it from proteolysis [157].

It is worth noting that angiotensin II is probably the main mediator of these pathogenic effects in both acute and chronic scenarios. Fortunately, both SIRT3 and SIRT6 exert compensatory mechanisms against it, mainly by easing angiotensin II-induced endothelial permeability (and hence proteinuria) and angiotensin II-induced fibrosis. The underlying mechanisms that mediate such effects are diverse and complex, but the stimulation of tight junction proteins such as Zonula occludens-1 and the downregulation of matrix metalloproteinases as well as the transforming growth factor beta pathway stand out [158,159].

Another effect of rising temperatures is crystal deposition due to low urine outputs. In this matter, SIRT3 has also been reported to inhibit calcium oxalate crystal formation and crystal–cell adherence by upregulating the nuclear factor-erythroid-2–related factor 2/HO-1 (HSP32) pathway and promoting M2 macrophage polarization via Forkhead Box O1 deacetylation, yielding a less inflammatory environment, which, in turn, leads to reduced apoptosis [160,161]. Furthermore, as reviewed earlier, uric acid is a key pathophysiological factor in AKI, particularly in the context of dehydration. In this regard, SIRT1 might have a role as it has been shown to be inhibited by hyperuricemia-related AKI alongside endothelial nitric oxide synthase [162].

#### 4.2.2. Sirtuins, Metabolism, and Adipose Tissue

Although compensatory mechanisms could bring damage under heat stress after chronic activation, it is worth noting that it is not the only course of action that causes damage at the molecular level. The absence of brown adipose tissue activation due to warmer temperatures could be one of the mechanisms in which predisposition for diabetic kidney disease takes place. Evidence has demonstrated that the prevalence of type 2 diabetes has been globally increasing in the past years. As diabetic kidney disease is a global leading cause of end-stage diabetic kidney disease, with uncontrolled glycemia being a main risk factor, global warming might be an important trigger behind the absence of brown adipose tissue activation due to warmer temperatures. Although reasons seem to be multifactorial, in the context of temperature stress, brown adipose tissue takes a more important role. When cold exposure takes place, the activity level of brown adipose tissue increases with the uptake of fatty acids to generate heat through lipid combustion [163,164,165]. This reduces the influx of fatty acids to other organs. Therefore, cold is an important stimulus for thermogenesis and brown adipose tissue utilization. However, when temperatures increase, brown adipose tissue mass and activity decreases, and this generates greater glycemic fluctuations as less glucose clearance takes place [166]. This correlates positively with rising global temperatures and greater diabetes incidence [167].

Brown adipose tissue is essential for dietary and adaptive thermogenesis, with β3 adrenergic response, cold exposure, and some nutrients such as fatty acids being the main stimuli for the mitochondrial uncoupling protein. The activity of this enzyme promotes not only thermogenesis, but also glucose metabolism and antioxidative activity in the kidney [168,169,170]. Brown adipose tissue activation has a renoprotective effect in diabetic mice through the secretion of beneficial adipokines, which antagonize angiotensin and activate the anti-inflammatory AMP-activated protein kinase/SIRTs/peroxisome proliferator-activated receptor gamma coactivator 1 alpha pathway [171]. In fact, low circulating levels of neuregulin 4, a brown adipose tissue-specific adipokine, has been associated with poorer metabolic and renal states in diabetic kidney disease murine models [172]. With rising temperatures, it is expected that brown adipose tissue differentiation will be less efficient, and its activation will be milder, consequently conditioning a deficient glucose metabolism and creating a favorable environment for the development of diabetic kidney disease.

Although no studies have directly sought an association between temperature stress, DKD, and sirtuins, there is enough evidence to suggest a potential positive role. Several members of the sirtuin family such as SIRT2, SIRT3, and SIRT6 have been shown to increase at cold temperatures and to improve brown adipose tissue-mediated thermogenesis by upregulating key effectors such as cAMP-response element binding protein and uncoupling protein 1 itself [173,174,175]. Of note, SIRT5 appears to have a central role in adipose tissue differentiation and browning, as demonstrated by the blockage of not only the master adipogenic transcription factors peroxisome proliferator-activated receptor gamma and CCAAT/enhancer binding protein alpha, but also the brown adipose tissue driver gene PR/SET Domain 16 in SIRT5 knockout models in multipotent mesenchymal cells [176]. Nonetheless, the exact role of SIRT5 in this process remains uncertain, as a more recent report found that pharmacologic inhibition of SIRT5, rather than SIRT5 expression, induces brown adipogenesis in preadipocytes in vitro [177]. These mechanisms are summarized in Figure 3.

Therefore, the various molecular interactions that take place through the SIRTs discussed above have beneficial roles against temperature-associated nephropathies. However, they possess a wide array of functions that mediate damage associated with various other nephropathies (Table 2). Thus, these proteins could become a potential therapeutic target to revert specific impairments associated with high temperatures in the kidneys.

## 5. Concluding Remarks and Future Perspectives

With the current knowledge, we know that extreme temperatures associated with weather change can induce organ damage and increase the susceptibility to the development of cardiovascular and renal diseases. Bearing in mind that climate change is not as simple as rising temperatures but is a complex process that involves extreme weather conditions, unfavorable rising temperatures, cold spells, and droughts, it is indispensable to set our foundation of knowledge about the molecular changes and physiological processes that take place in organisms under these conditions. Climate change is becoming a newly established risk factor for a variety of diseases; for instance, the proposed concept of the COCCI (Cardiovascular diseases, Obesity, Climate Change, Inflammation) syndemic is gaining recognition [224]. In this sense, the kidney, a key organ in maintaining electrolyte and volume balance in states of stress, is also significantly impacted by acute rising temperatures. Therefore, temperature change (specifically heat-induced stress) represents an important trigger for renal diseases.

As we have thoroughly analyzed, kidneys are essential for thermal adaptation and possess compensatory mechanisms to adjust to such thermic changes and volume losses. Chronic activation of counterregulatory mechanisms against heat stress elicited by the kidney can lead to maladaptive processes that exert further kidney injury [9]. Therefore, these mechanisms are beneficial up to a certain threshold after which they do more harm than good. These effects prove to be the link between climate change and the alarmingly rising incidence of acute and chronic kidney diseases in recent years, especially in tropical regions. In recent years, protein families such as phosphofructokinase, HSPs, and sirtuins have been identified as key regulators of the overactivation of these physiological responses due to their pro- and anti-inflammatory, antioxidative, and metabolism-optimizing functions. Therefore, researching the underlying mechanisms represents an area of opportunity for the identification of new biomarkers and therapeutic targets.

The importance of the topic of climate change is intertwined with public health regardless of economic stratum as renal events seem to be affecting a diverse population from different socioeconomic backgrounds. Research on temperature and renal diseases have expanded from concentrating in tropical regions and agricultural workers and miners to now include hospitalized patients in Europe, Australia, and North America [37,225,226]. In these regions, the use of centralized air-conditioning might be more predominant to combat temperature extremes. However, in the long run it prevents acclimatization when exposed to high temperatures, and the compensatory mechanisms described above will still be activated under stress conditions. In cold spells, our knowledge of renal homeostasis is limited but still behaves as a stressor. Laying the evidence described above into perspective, this phenomenon seems to affect individuals regardless of occupation and standard of living.

Studies about cardiovascular risk and climate change have thoroughly compared both extremes in temperature bringing interesting conclusions, feeding our knowledge about heat and cold effects in the development of diseases and how the effects are unique and separate. However, on kidneys, such comparative studies do not exist, and the effects of cold spells continue to remain greatly unexplored. Even though past and present studies in regard to weather change and renal diseases situate heavily in heat stroke and warmer temperatures, it will be interesting to open the frontiers of research in regard to colder temperatures.

## Figures and Tables

**Figure 1 biomolecules-14-00251-f001:**
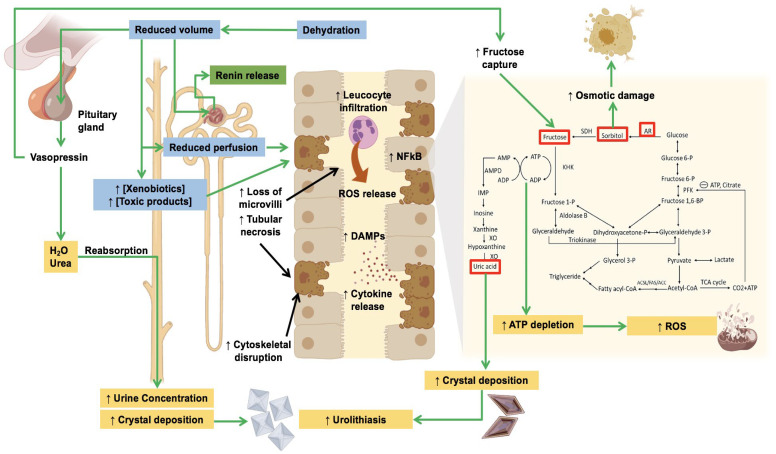
Compensatory mechanisms associated with extreme temperatures and their detrimental chronic effects on tubular and glomerular cells. Abbreviations: AR: aldose reductase; ATP: adenosine triphosphate; H_2_O: water; NF-*k*B: nuclear factor kappa-beta; ROS: reactive oxygen species; the brackets [] flanking a substance refer to the concept “concentration” of the substances inside it. Upregulated enzymes in the metabolic shift produced by vasopressin are marked with red rectangles.

**Figure 2 biomolecules-14-00251-f002:**
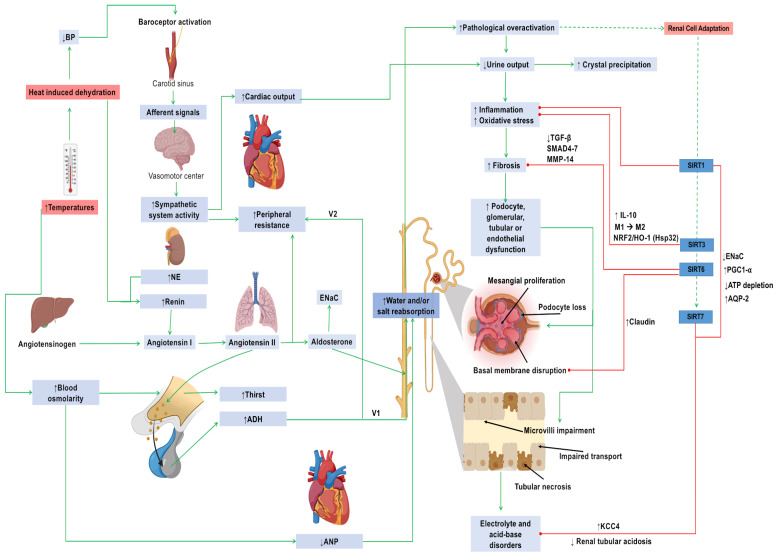
Acute compensatory response elicited by sirtuins against heat-induced stress. Abbreviations: ADH: antidiuretic hormone; AQP-2: aquaporin 2; BP: blood pressure; IL: interleukin; MMP: metalloproteinase; NE: norepinephrine; SIRT: sirtuins; V1: V1 receptor; V2: V2 receptor. Green arrows indicate induction, while red arrows indicate inhibition. Upward arrows indicate upregulation, while downward arrows indicate downregulation.

**Figure 3 biomolecules-14-00251-f003:**
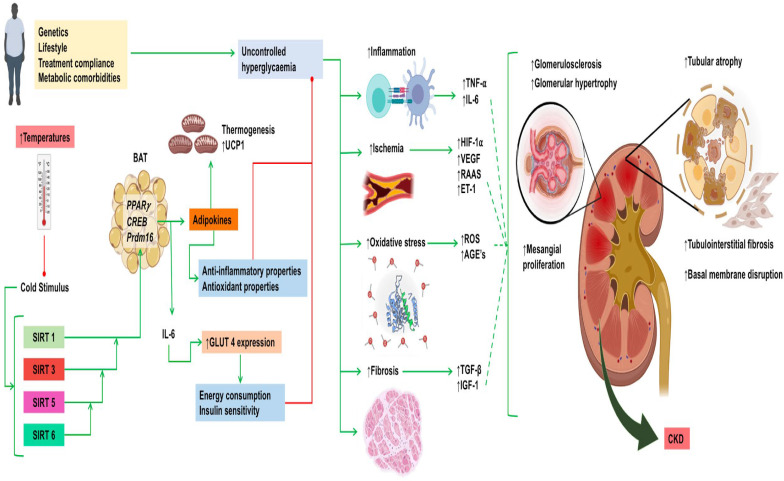
Chronic compensatory response elicited by sirtuins against heat-induced stress. Abbreviations: AGE’s: advanced glycation end products; BAT: brown adipose tissue; CKD: chronic kidney disease; ET-1: endothelin-1; HIF-1α: hypoxia inducible factor 1-alpha; IL: interleukin; RAAS: renin–angiotensin–aldosterone axis; ROS: reactive oxygen species; SIRT: sirtuins; TNF-α: tumor necrosis factor-alpha; UCP-1: uncoupling protein-1; VEGF: vascular endothelial growth factor. Green arrows indicate induction, while red arrows indicate inhibition.

**Table 1 biomolecules-14-00251-t001:** The main mechanisms exerted by heat shock protein families in the protective response against heatstroke in the renal system.

HSP	Nephron Location	Heat Stress Pathophysiological Pathways Involved	Main Renal Protective Functions	References
HSP27	PCT and medullary (distal) CT	AKI due to dehydration or drug toxicity (cyclophilin).	↓ TNF-α injury, ↓ Bad, Bax, and caspase activation, ↓ osmotic damage, ↑ podocyte barrier function, ↑ glutathione bioavailability, ↓ ROS, ↑ cytoskeleton maintenance	[101,102,103,104,107,108,148]
HSP32	Mesangium, cortical PCT	Heavy metal intoxication, AKI due to dehydration or ischemia, or oxidative stress.	↓ IL-1, IL-6, and IL-8, ↓ ROS bioavailability, ↓ TGF beta	[137,138,139,140,141,142,143,144,145,146]
HSP60	Corticomedullary PCT and Henle’s loop	AKI due to dehydration, hypoperfusion, osmotic damage, or heavy metal toxicity.	↓ Tubular necrosis, cytoskeleton repairing, and mitochondrial protein re-folding	[109,129,131,133,134,135,136]
HSP70	PCT, medullary CT, Henle’s loop, podocyte slits	AKI due to dehydration and ischemia.	↓ Bax and caspase 3, ↑ preservation of Na^+^-K^+^-ATPase and electrolyte transport maintenance, ↓ NF-*k*B p65 translocation, ↓ TGF-β, ↓ COL1 deposition, ↑ ERK	[99,100,111,114,115,116,118,149]
HSP90	Bowman’s space, Mesangium, medullary Henle’s loop	AKI due to dehydration and ischemia or nephrolithiasis.	↑ Kidney function due to ↑ PKM2-Akt, ↓caspase activity, ↑ autophagy by ↑ HIF-1α-BNIP3/BNIP3L, ↓ β-actin, vimentin, and calpain-1 oxalate binding, ↓TLR4 and TNF-α	[120,121,122,123,125,127,128,129]

Abbreviations: AKI, acute kidney injury; COL1, collagen 1; CT, collector tubule; ERK, extracellular-signal-regulated kinase; HSPs, heat shock proteins; NF-*k*B, nuclear factor kappa-B; PCT, proximal convoluted tubule; ROS, reactive oxygen species; TGF-β, transforming growth factor beta; TLR4, Toll-like receptor 4; TNF-α, tumor necrosis factor alpha. ↑ Upward arrows express upregulation, ↓ downward arrows indicate inhibition.

**Table 2 biomolecules-14-00251-t002:** Main mechanisms of sirtuins in kidney health and disease.

Class	SIRT	Main Activity	Nephropathy	Molecular Mechanisms	Reference
I	1	Deacylase	DKD	↑ Mitochondrial function and mitophagy, ↑ (PGRN-PGC1α/PPARɣ) ↓ Inflammation, ↓ (NF-*k*B, STAT3) ↓ Apoptosis, ↓ (caspase 3, cleaved PARP), ↑ mTOR ↓ Fibrosis (Keap1/Nrf2, ARE), ↓ (Notch, Timp-1, FN, and TGF-β)	[178,179,180,181,182,183,184,185,186]
			Sepsis-related AKI	↓ Inflammation, ↓(HMGB-1, NLRP3 inflammasome, IL18, and IL-1β)	[187,188]
			Tubulointerstitial fibrosis	↓ Fibrosis, ↓ (HIF-1α, HIF-2α, TGF-β, and ECM genes)	[189,190]
			Toxic AKI (heavy metals)	↓ Pyroptosis, ↓ (XBP-1/IRE-1α/NLRP3 inflammasome)	[191]
			Glomerulosclerosis	↓ Mesangial proliferation and fibrosis, ↓ (COL4, α-SMA, and TGF-β) ↓ Oxidative stress, ↑ (Nrf2/ARE, SOD1, and HO-1)	[183,192]
			Ischemia-reperfusion	↑ Autophagy, ↑ (LC3B-II, beclin1)	[193]
			Renal cell carcinoma	↓ Tumorigenesis, ↓ STAT3/FGB ↑ Tumorigenesis, ↓ (p53, p21/Cip1)	[194,195]
	2	Deacylase	Tubulointerstitial fibrosis	↑ Fibrosis, ↑ TGF-β1/MDM2	[196]
			Ischemia-reperfusion	↑ Apoptosis, ↑ FOXO3α/FasL	[197]
			AKI [cisplatin]	↑ Apoptosis and necroptosis, ↑ (MKP-1, JNK)	[198]
	3	Deacylase	DKD	↓ Fibrosis, ↓ (HIF-1α, TGF-β/Smad3/4, and aberrant glycolysis) ↓ Apoptosis, ↑ Akt/FoxO ↓ Oxidative stress, ↑ (IDH2, SOD2)	[199,200,201]
			Toxic AKI (cisplatin)	↓ Oxidative stress, ↑ (Nrf2/SOD2, GPx, and catalase) ↑ FAO, ↑ LKB1/AMPK	[202]
			Nephrolithiasis	↓ Apoptosis, ↓Bax ↑ Crystal-cell adherence, ↑ NRF2/HO-1	[160,161]
			Hypertensive nephropathy	↓ Endothelial-to-mesenchymal transition ↑ FOXO3α/catalase ↓ Fibrosis, ↓ (KLF15, FN, and COL4)	[196,203,204,205]
			Ischemia-reperfusion	↑ Mitochondrial fusion, ↑ ERK/OPA1 ↓ Apoptosis, ↑ Bcl-2/c-IAP, ↓ (cytochrome c, caspase 9) ↓ Inflammation, ↓ (MCP1, IL-6, and TNF-α) ↓ Oxidative stress markers: GSSG, MDA, and LPO, ↑ (SOD, GPX)	[206]
			Renal cell carcinoma	Warburg effect reversal ↑ Mitochondriogenesis, ↑ TFAM	[207]
			DKD	↓ Inflammation, ↓ (IL-1β, IL-6, and TNF-α) ↓ Apoptosis, ↑ Bcl-2, ↓ (NOX1, Bax, and p38)	[208]
II	4	Ribosyl-transferase	Renal cell carcinoma	Potential role in chemoresistance and metastasis	[209]
			Toxic AKI (cisplatin)	↓ Apoptosis, ↑ Bcl-2, ↓ (caspase 3, cleaved PARP, and cytochrome *c*) ↓ Oxidative stress, ↑ Nrf2/HO-1 ↓ Mitochondrial fission, ↓ Drp1	[210,211]
III	5	Deacylase	Ischemia-reperfusion	Mitochondrial vs. peroxisomal FAO ratio dysregulation	[212]
			Renal cell carcinoma	↑ Mitochondrial O_2_ consumption, ↑ GDH ↑ Tumorigenesis, ↓ SDH5	[213,214]
			Various podocytopathies	↓ Inflammation, ↓ (IL-1β, IL-6, TNF-α, and Notch) ↑ Autophagy, ↑ LC3B-II	[215]
IV	6	Ribosyl-transferase Deacylase	Hypertensive nephropathy	↓ Endothelial cell permeability, ↓ (MMPs, WT1, and GATA5), ↑ ZO-1 ↓ Endothelial senescence, ↓b-galactosidase ↑ Autophagy, ↑ LC3B-II ↑ Cholesterol efflux, ↑ ABCG1 ↓ Fibrosis, ↓ Angiopoietins/TGF-β	[158,159,216]
			Sepsis-related AKI	↓ Inflammation: IL-6 and TNF-α ↑ Autophagy, ↑ LC3B-II	[217]
			Tubulointerstitial fibrosis	↓ Fibrosis, ↓ (Wnt/β -catenin/TGF-β, GSK3β)	[218]
			DKD	↑ Mitochondrial function, ↑ AMPK ↓ Inflammation, ↑ M2 macrophage transformation	[219,220]
			Toxic AKI (cisplatin)	↓ Inflammation and apoptosis, ↓ (NF-*k*B, ERK1/2)	[221]
			DKD	↓ Apoptosis, ↓ Caspase 3	[222]
	7	Deacylase	Toxic AKI (cisplatin)	↑ Inflammation: NF-*k*B, TNF-α	[223]

Green: protective; yellow: uncertain; red: pathogenic. ↑ Upward arrows express upregulation; ↓ downward arrows indicate inhibition. ABCG1, ATP-binding cassette sub-family G member 1; AKI, acute kidney injury; AMPK, 5′AMP-activated protein kinase; ARE, antioxidant response elements; BAX, Bcl-2-associated X protein; Bcl-2, B-cell lymphoma 2 protein; COL4, collagen 4; DKD, diabetic kidney disease; Drp1, dynamin-1-like protein; ECM, extracellular matrix; ERK, extracellular-signal-regulated kinase; FAO, fatty acid oxidation; FasL, Fas ligand; FGB, fibrinogen beta chain; FN, fibronectin; FOXO3, forkhead box O3; GDH, glutamate dehydrogenase; GPx, glutathione peroxidase; GSK3β, glycogen synthase kinase 3 bet; GSSG, glutathione disulfide; HIF, hypoxia inducible factor; HMGB-1, high mobility group box 1 protein; HO-1, heme oxygenase 1; IAP, inhibitor of apoptosis; IDH, isocitrate dehydrogenase; IL, interleukin; IRE-1α, inositol-requiring transmembrane kinase/endoribonuclease 1 alpha; JNK, c-Jun N terminal kinases; Keap1, Kelch-like ECH-associated protein 1; KLF15, Krüppel-like factor 15; LC3B-II, microtubule-associated protein 1A/1B light chain 3; LKB1, liver kinase B1; LPO, lipid peroxidation; MCP1, monocyte chemoattractant protein 1; MDA, malondialdehyde; MDM2, mouse double minute 2 homolog; MKP-1, mitogen-activated protein kinase phosphatase 1; MMP, matrix metalloproteinases; mTOR, mechanistic target of rapamycin; NF-*k*B, nuclear factor kappa-light-chain-enhancer of activated B cells; NLRP3, NOD-, LRR-, and pyrin domain-containing protein 3; NOX1, NADPH oxidase 1; Nrf2, nuclear factor-erythroid factor 2-related factor 2; OPA1, optic atrophy 1 protein; p21/Cip1, cyclin-dependent kinase inhibitor 1; PARP, poly (ADP-ribose) polymerase; PGC1α, peroxisome proliferator-activated receptor co-activator-1 alpha; PGRN, progranulin; PPARɣ, peroxisome proliferator-activated receptor gamma; SDH, succinate dehydrogenase; SOD, superoxide dismutase; STAT3, signal transducer and activator of transcription 3; TFAM, mitochondrial transcription factor A; TGF-β, transforming growth factor beta; Timp1, TIMP metallopeptidase inhibitor; TNF-α, tumor necrosis factor alpha; WT1, Wilms tumor 1; XBP1, X-box binding protein 1; ZO-1, Zonula occludens 1; α-SMA, alpha smooth muscle actin.

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
