# Peer review of "Molecular Challenges and Opportunities in Climate Change-Induced Kidney Diseases"

_biomolecules, 2024, doi:10.3390/biom14030251_

Round 1

Reviewer 1 Report (Previous Reviewer 2)

Comments and Suggestions for Authors

A comprehensive review of the kidney during the changes of climate. 

Perhaps a more adequate title for this review would be to stress climate changes instead of temperature.

There are only minor issues that can be found in the paper:

line 94 kidney stone formation

line 108 on kidney function (not in kidneys)

line 228 angiotensin II and aldosterone levels/concentrations but not activity

Author Response

Response to the reviewers’ comments

We thank the reviewers for their time and their valuable feedback on our manuscript. We have revised the manuscript in accordance with their suggestions. We present a point-by-point response below and added or removed any text vis-à-vis the comments in the manuscript text in BLUE font.

Reviewer #1

Comments 1:  A comprehensive review of the kidney during the changes of climate. 

Perhaps a more adequate title for this review would be to stress climate changes instead of temperature.

There are only minor issues that can be found in the paper:

Thank you for your comments.

Response: Thanks for your recognition. Following your recommendations we have changed the title as follow:

“Molecular challenges and opportunities in climate change-induced kidney diseases”

Comment 2: line 94 kidney stone formation.

Response: Thank you for your comments. When we mentioned about the U.S. kidney stone "belt" it refers to “the region in the southeastern United States where the rate of kidney stones, or kidney calculi, is excessive”. It is implicit kidney stone formation, reason we decided not to add kidney stone formation.

Comment 3: line 108 on kidney function (not in kidneys).

Response: Thank you for your observation. We changed it on the manuscript.

Comments 4: line 228 angiotensin II and aldosterone levels/concentrations but not activity.

Response: We appreciate your observation. We have changed it as follows:

“It has been long observed that plasma renin, angiotensin II, and aldosterone levels rise upon environmental heat stress…”. 

Reviewer 2 Report (New Reviewer)

Comments and Suggestions for Authors

Overall, the manuscript is well written and appears to be an appropriate overview of the literature. The introduction is brief, but appropriate for setting the stage for what the review wishes to focus on. Sections 2 and 3 seem to be included primarily for the sake of providing the reader with sufficient information on kidney processes to understand the mechanisms of heat shock proteins and sirtuins. Section 4 is the focus of the manuscript and is a thorough dive into the mechanisms of HSP/SIRTs under various thermal stresses. There are some minor revisions that would improve the flow and clarity of this manuscript prior to publication.

Title: Title could be reworked. What is meant by ‘molecular challenges’?

Line 11: I recommend rewording “affect cardiovascular health, especially the kidneys”. The current sentence suggests that kidneys are a part of the cardiovascular system rather than being negatively impacted by deleterious changes to the cardiovascular system.

Lines 72-75: It is unclear if authors are suggesting these posited factors are directly caused by climate change or are merely being exacerbated by climate change. The cited reference (#23) appears to suggest that such factors are exacerbated by climate change, which is an important distinction. Clarifying this statement would be beneficial for reader understanding.

Lines 75-80: While there is a survey based study that has found evidence of decreased kidney function among miners (https://doi.org/10.1053/j.ajkd.2009.12.012), it does not address causality. If the authors are suggesting this is due to heat stress, an additional citation would be beneficial. There are other occupations (such as fishing) which experience comparable heat exposure, but do not demonstrate increased rates of CKDu (https://doi.org/10.1016/j.joclim.2022.100143). There is certainly enough evidence in farming to suggest a relationship between heat exposure and CKDu, but this does not seem to be the case in mining.

Line 124: Recommend replacing “non-sensible” with insensible if this is referring to the mechanism of insensible vs sensible fluid loss.

Line 363: Consistent should be used rather than consistently.

Line 414: “Although to a lesser extent” seems to be referring to a previous section rather than the sentence which contains it. This could likely be removed.

Line 444: “mitochondrial rial proteins” appears to be a typo.

Line 568: “Trigger behind”? It is unclear to what this refers. Behind what?

Line 649: Conclusion section appears to be mistakenly labeled as section 4 rather than section 5.

Lines 650-657: These two sentences should be rewritten. Both are unnecessarily long and convoluted without making a clear statement.

Throughout the manuscript, authors state and restate what previous sections have discussed and what upcoming sections will discuss. This adds unnecessary length and disrupts overall flow. Recommend editing intro and outro of sections to improve succinctness.

Comments on the Quality of English Language

I suggest some minor editing as noted in my general comments to the authors.  

Author Response

Reviewer #2

Comment 1: Overall, the manuscript is well written and appears to be an appropriate overview of the literature. The introduction is brief, but appropriate for setting the stage for what the review wishes to focus on. Sections 2 and 3 seem to be included primarily for the sake of providing the reader with sufficient information on kidney processes to understand the mechanisms of heat shock proteins and sirtuins. Section 4 is the focus of the manuscript and is a thorough dive into the mechanisms of HSP/SIRTs under various thermal stresses. There are some minor revisions that would improve the flow and clarity of this manuscript prior to publication.

 Title: Title could be reworked. What is meant by ‘molecular challenges’?

Response: We thank the reviewer for pointing this comment. When we speak about challenge or molecular challenges in our title, we relate this words as the Cambridge dictionary defines “challenge”: “(the situation of being faced with) something that needs great mental or physical effort in order to be done successfully and therefore tests a person's ability”, in our case focusing on the molecules in the kidney that need to be modulated, over/down expressed, or to be regulated in order to prevent a kidney damage.

We think this noun “challenge” explains or integrates our main idea of our work. Keeping this in mind, but following both reviewers’ comments, we reworked our title to strengthen our idea. It will be as follow:

“Molecular challenges and opportunities in climate change-induced kidney diseases”

Comment 2: Line 11: I recommend rewording “affect cardiovascular health, especially the kidneys”. The current sentence suggests that kidneys are a part of the cardiovascular system rather than being negatively impacted by deleterious changes to the cardiovascular system.

Response: We appreciate your suggestion to rework this phrase. We modified it as follow:

“Physiological distress due to low and high temperatures can affect the heart, blood vessels, liver, and especially, the kidneys.”

Comment 3: Lines 72-75: It is unclear if authors are suggesting these posited factors are directly caused by climate change or are merely being exacerbated by climate change. The cited reference (#23) appears to suggest that such factors are exacerbated by climate change, which is an important distinction. Clarifying this statement would be beneficial for reader understanding.

Response: To avoid any confusion, we rephrase this sentence, as follow:

“As its name implies, its pathophysiology is not well known and its exact source is uncertain; however, it is widely accepted that heat stress, dehydration, strenuous labor, use of agrochemicals, exposure to heavy metals, and water pollution are contributing factors or as Priyadarshani et al. (2022), have mentioned, as “suspected causative factors” for this pathology [23].”

Comment 4: Lines 75-80: While there is a survey based study that has found evidence of decreased kidney function among miners (https://doi.org/10.1053/j.ajkd.2009.12.012), it does not address causality. If the authors are suggesting this is due to heat stress, an additional citation would be beneficial. There are other occupations (such as fishing) which experience comparable heat exposure, but do not demonstrate increased rates of CKDu (https://doi.org/10.1016/j.joclim.2022.100143). There is certainly enough evidence in farming to suggest a relationship between heat exposure and CKDu, but this does not seem to be the case in mining.

Response: We agree with the reviewers’ comments and observations. To avoid any confusion or misunderstood, we rephrase the sentence as follow:

“It is not surprising that mine and farm workers present a greater decrease in their kidney function due to their labor conditions [24-26], nevertheless, it cannot be addressing causality to only one or other factors as it was mentioned before, If not to the stress conditions given to the individual at a certain moment in their occupational labor.”

Comment 5: Line 124: Recommend replacing “non-sensible” with insensible if this is referring to the mechanism of insensible vs sensible fluid loss.

Response: Thank you for the appreciation. We have changed the word in the manuscript.

“Acute or chronic exposure to high temperatures leads to insensible volume losses”

Comment 6: Line 363: Consistent should be used rather than consistently.

Response: We have modified it on the manuscript.

Comment 7: Line 414: “Although to a lesser extent” seems to be referring to a previous section rather than the sentence which contains it. This could likely be removed.

Response: We thank the reviewer for this observation. We removed it from the line.

“HSP60 has also been associated to exert protective effects within the nephron that can play a critical role in the defense of heatstroke”

Comment 8: Line 444: “mitochondrial rial proteins” appears to be a typo.

Response: We removed the typo.

Comment 9: Line 568: “Trigger behind”? It is unclear to what this refers. Behind what?

Response: We appreciate your observation. We completed the phrase as follow:

“As diabetic kidney disease is a leading cause of end-stage diabetic kidney disease globally, with uncontrolled glycemia being a main risk factor, global warming might be an important trigger behind the absence of brown adipose tissue activation due to warmer temperatures”.

Comment 10: Line 649: Conclusion section appears to be mistakenly labeled as section 4 rather than section 5.

Response: Thank you. We changed the numbering.

Comment 11: Lines 650-657: These two sentences should be rewritten. Both are unnecessarily long and convoluted without making a clear statement.

Response: We rewrite the sentences, removing unnecessary information as follow:

“With current knowledge, we know that extreme temperatures associated with weather change can induce organ damage and increase the susceptibility to the development of cardiovascular and renal diseases. Bearing in mind that climate change is not as simple as rising temperatures but a complex process that involves extreme weather conditions, unfavorable rising temperatures, cold spells, and droughts, it is indispensable to set our foundation of knowledge about molecular changes and physiological processes that take place in our organism under these conditions. Climate change is becoming a newly established risk factor for a variety of diseases; for instance, the proposed concept of the COCCI (Cardiovascular diseases, Obesity, Climate Change, Inflammation) syndemic is gaining recognition [224]. In this sense, the kidney, a key organ in maintaining electrolyte and volume balance in states of stress, is also significantly impacted by acute rising temperatures. Therefore, temperature change (specifically heat-induced stress) represents an important trigger for renal diseases.”

Comments 12: Throughout the manuscript, authors state and restate what previous sections have discussed and what upcoming sections will discuss. This adds unnecessary length and disrupts overall flow. Recommend editing intro and outro of sections to improve succinctness.

Response: We thank the reviewer for pointing this fruitful suggestion. We removed some of those unnecessary intro and outro sections, for example:

“Further elucidation of these markers will be provided in the subsequent section”, “As previously mentioned, another group of proteins, namely sirtuins, has demonstrated comparable effects to the HSP families concerning the protection of renal cells against stress. This will be further discussed in the following section”, “The following sections will focus on how SIRTs can elicit protective mechanisms in the disarrangement of RAAS system and the metabolic changes associated with exposure to adverse temperatures”.

Comments 13: Comments on the Quality of English Language

I suggest some minor editing as noted in my general comments to the authors.  

Response: Thank you for your observations and recommendations that improved our manuscript.

Reviewer 3 Report (New Reviewer)

Comments and Suggestions for Authors

This manuscript covers a very interesting subject in great detail.

It is didactic and easy to read.

The subject is covered comprehensively 

Author Response

Thanks for your recognition, the authors appreciate the appraisal.

This manuscript is a resubmission of an earlier submission. The following is a list of the peer review reports and author responses from that submission.

Round 1

Reviewer 1 Report

Comments and Suggestions for Authors

 Comments on the manuscript titled: “Molecular challenges and opportunities in temperature-induced kidney diseases“ by Eder Luna-Cerón et al. submitted to mdpi-Biomolecules

The manuscript is a very valuable and comprehensive review paper which discusses several molecules that play a crucial role in preventing or provoking cellular damage in extreme temperatures, including vasopressin, aldose reductase, adenosine triphosphate (ATP), nuclear factor kappa-beta (NFkB), and reactive oxygen species (ROS).

Based on the multiple literature reports (studies from various years (1988-2022, mostly from period 2015-2019) and research groups), the Authors show the evidence that dehydration can lead to a decrease in blood flow to the kidneys, which can trigger stress on cells in the kidneys.

The generated diagrams and illustrations deserve appreciation, as they are a very helpful source of detailed information in a compact form.

The article fails to directly address whether extreme temperature conditions can be averted from triggering the activation of the renin-angiotensin system. Nevertheless, it delves into the renin-angiotensin-aldosterone system (RAAS) and its correlation with hypertension, a condition linked to seasonal fluctuations. Research indicates elevated blood pressure readings in colder climates compared to warmer ones, with potential links to the mechanisms of the renin-angiotensin system.

The discussion centers on the physiological distress triggered by extreme temperatures, impacting cardiovascular health, notably the kidneys. The reduction in circulating plasma volume through sweat induces stress on the renal and cardiovascular systems, associated with certain molecules pivotal in preventing or inciting cellular damage. The authors propose that as temperatures fluctuate due to changing weather patterns, an increasing number of regions face exposure to extreme heat and cold, potentially resulting in temperature-induced kidney diseases.

As the article is a review paper based on information from data taken from other already published sources and does not have a separate methodology, the focus can only be on the aspect of presenting the facts and results in a transparent manner. 

Please pay attention to the numbering and formatting of the chapters and subchapters. Is chapter 1.1. belonging to Introduction or should it function as a separate chapter. Change numbering or font accordingly.

Line 54: - on young agricultural workers- reported   …. Re-write whole sentence

Figure 1 caption: what “substances inside brackets represent concentration” really means?

The sentence "Although the focus in some of these reports has been cardiovascular diseases to establish a clear association with cardiovascular mortality, there are no studies exploring winter-associated blood pressure rise and CKD in the long run" could benefit from rephrasing for clarity and coherence.

In line 208  there is a mention of "beneficial key biomarkers" without further explanation. It could be helpful to provide context or specify what these biomarkers are and how they relate to mitigating kidney damage caused by extreme temperatures.

The paragraph 3 introduces heat shock proteins (HSPs) and their role in cellular protection against stress, particularly heat stress. However, it shifts to discussing global warming's potential impact on HSP expression without a clear transition or connection to the initial focus on cellular stress resistance. Connecting these concepts more cohesively could enhance understanding.

The statement from lines 226-228 "Although it is not completely elucidated, the kidney can activate compensatory mechanisms to afront the detrimental effects associated with exposure to high temperatures" lacks specificity and might benefit from further clarification or supporting evidence.

The Authors mention the dual role of HSPs in both enhancing stress tolerance and potentially causing cell cycle arrest or cell death. However, it suggests limited evidence supporting this dual role, indicating the need for further research. Clarity could be improved by emphasizing the need for additional investigation to ascertain the true consequences of increased HSP levels, especially in the context of climate change.

Line 419 please re-write sentence and do not start with “Another…

Please check whether the  paragraphs have a clear structure and transitions between different ideas, as in the present form the article makes it challenging to follow the logical flow of the arguments. Breaking it into smaller, more focused paragraphs could improve readability. The text could benefit from smoother transitions between different concepts and ideas.

Tables should be formatted according to the journal requirements.

References should be formatted according to the journal requirements. Example: Author 1, A.B.; Author 2, C.D. Title of the article. Abbreviated Journal Name Year, Volume, page range (comma after surname, full stop after initial, semi-colon between)

Year, pages and other details missing in refs 8, 18; remove [date unknown] from ref 30; remove Chinese signs from ref 47; check if all authors needed in ref 58

Good Job! Good Luck to Authors!

Comments on the Quality of English Language

Minor corrections required

Reviewer 2 Report

Comments and Suggestions for Authors

The mechanisms are well described, but, as we are mammalians characterized by strict thermoregulation. The changes in body temperature are restricted mostly to the skin and lower extremities. Body heat is mostly generated by the liver and skeletal muscles. The kidney's role in the generation of heat is minor, and mostly related to their function. The location of the kidney precludes changes in their temperature, except in the extremal situation. When the temperature of the surroundings is greater than that of the skin, the only means by which the body can rid itself of heat is evaporation. This generates excessive thirst and the need for increased fluid intake - which prevents hypovolemia.

Therefore the hypothesis of the paper is elusive and exaggerated.